# Isolation and Functional Characterization of Two *CONSTANS-like 16* (*MiCOL16*) Genes from Mango

**DOI:** 10.3390/ijms23063075

**Published:** 2022-03-12

**Authors:** Yuan Liu, Cong Luo, Yihang Guo, Rongzhen Liang, Haixia Yu, Shuquan Chen, Xiao Mo, Xiaozhou Yang, Xinhua He

**Affiliations:** State Key Laboratory for Conservation and Utilization of Subtropical Agro-Bioresources, College of Agriculture, National Demonstration Center for Experimental Plant Science Education, Guangxi University, Nanning 530004, China; 18638785643@163.com (Y.L.); 22003luocong@gxu.edu.cn (C.L.); yihang1066@163.com (Y.G.); liangrz0207@163.com (R.L.); yuhaixia0201@163.com (H.Y.); shuquanchen2022@163.com (S.C.); 18376683951@163.com (X.M.); yxz05206018@163.com (X.Y.)

**Keywords:** mango, *CONSTANS*, flowering, functional analysis, abiotic stress

## Abstract

*CONSTANS* (*CO*) is an important regulator of photoperiodic flowering and functions at a key position in the flowering regulatory network. Here, two *CO* homologs, *MiCOL16A* and *MiCOL16B*, were isolated from “SiJiMi” mango to elucidate the mechanisms controlling mango flowering. The *MiCOL16A* and *MiCOL16B* genes were highly expressed in the leaves and expressed at low levels in the buds and flowers. The expression levels of *MiCOL16A* and *MiCOL16B* increased during the flowering induction period but decreased during the flower organ development and flowering periods. The *MiCOL16A* gene was expressed in accordance with the circadian rhythm, and *MiCOL16B* expression was affected by diurnal variation, albeit not regularly. Both the MiCOL16A and MiCOL16B proteins were localized in the nucleus of cells and exerted transcriptional activity through their MR domains in yeast. Overexpression of both the *MiCOL16A* and *MiCOL16B* genes significantly repressed flowering in *Arabidopsis* under short-day (SD) and long-day (LD) conditions because they repressed the expression of *AtFT* and *AtSOC1*. This research also revealed that overexpression of *MiCOL16A* and *MiCOL16B* improved the salt and drought tolerance of *Arabidopsis*, conferring longer roots and higher survival rates to overexpression lines under drought and salt stress. Together, our results demonstrated that *MiCOL16A* and *MiCOL16B* not only regulate flowering but also play a role in the abiotic stress response in mango.

## 1. Introduction

In higher plants, floral transition is the process that describes the switch from the vegetative stage to the reproductive stage. The time for this process is referred to as flowering time. The flowering mechanism of the annual plant species *Arabidopsis* is thoroughly understood. According to recent research, the onset of flowering is regulated in a timely manner by an intricate network involving a series of regulatory pathways, such as gibberellin, photoperiod, autonomous, aging and vernalization pathways [1,2,3]. Of the many regulatory pathways, the photoperiod pathway is especially important; it is involved in plant responses to photoperiod sensing and subsequent molecular events [4].

Many photoperiod pathway-related genes have been discovered, such as TIME OF CAB EXPRESSION 1 (TOC1), LATE ELONGATED HYPOCOTYL (LHY), EARLY FLOWERING 4 (ELF4), GIGANTEA (GI), CIRCADIAN CLOCK ASSOCIATED 1 (CCA1), FLOWERING LOCUS T (FT) and CONSTANS (CO) [5]. Among them, CO is a key component of which there are orthologs in various plant species [6,7,8,9]. Currently, CYCLING DOF FACTOR (CDF) is known as the only transcription factor that directly binds to the CO promoter and suppresses the expression of CO [10,11,12]. The GI gene plays a key role in the photoperiod induction pathway and positively regulates the expression of the CO gene under long-day conditions [13]. Overexpression of the FLOWERING BHLH (FBH) gene elevates CO levels without being affected by photoperiod [14]. In addition, phytochrome-interacting factors (PIFs) interact with CO to suppress flowering [15]. The E3 ubiquitin ligase-encoding gene HIGH EXPRESSION OF OSMOTICALLY RESPONSIVE GENES1 (HOS1) is involved in controlling the abundance of CO, and CONSTITUTIVE PHOTOMORPHOGENIC1 (COP1), as a flowering repressor, is regulated by cryptochromes (CRY) and promotes the proteolysis of CO in the dark [16,17]. CO genes belong to the BBX family and can be divided into three categories according to their domains in Arabidopsis [7,18]. Group I genes have one CO, CO-like, and TOC1 (CCT) domain and two B-box domains; group II genes have one B-box and one CCT domain; and group III genes have one B-box, one variant B-box and one CCT domain.

However, the functions of *CO* orthologs vary across different species. In *Arabidopsis*, overexpression of the *AtCO, AtCOL5* and *AtCOL16* genes promotes flowering under long-day (LD) or short-day (SD) conditions [19,20], but *AtCOL7*, *AtCOL8* and *AtCOL9* inhibit flowering under LD conditions [21,22]. In rice, the *OsHd1* gene delays flowering under LD conditions and promotes flowering under SD conditions [23], and *OsCOL16* inhibits flowering under both SD and LD conditions [24]. *StCO*, a *CO* homolog in potato, regulates flowering [25]. Moreover, the *CO* orthologs in Fuji apple, *MdCOL1* and *MdCOL2*, play a significant role in the growth and development of reproductive organs [26]. Thus, homologous *CO* genes have a wide range of effects on plant flowering and development.

Mango (*Mangifera indica* L.) is a member of the Anacardiaceae family and is an economically important fruit tree species. Flowering time has a considerable influence on the yield and quality of mango. Therefore, the discovery and identification of flowering-related genes are necessary for mango production. Mango flowering is the result of a complex process influenced by many factors, but it is not affected by daylength [27]. Several flowering-related regulatory genes have recently been identified in mango. For example, the *MiFT*, *MiSOC1* and *MiAP1* genes promote flowering, but the *MiCO* gene inhibits flowering in *Arabidopsis* [28,29,30,31]. However, the *COL16* gene has not been studied in mango. Here, we isolated two *COL* genes, *MiCOL16A* and *MiCOL16B*, and verified their function. Our results indicated that these two genes strongly influence the flowering and abiotic stress responses of mango.

## 2. Results

### 2.1. Isolation and Analysis of MiCOL16A and MiCOL16B

Two *CO* homologs, *MiCOL16A* (GenBank No: MW326761) and *MiCOL16B* (GenBank No: MW326762), were identified from *M. indica.* L. cv. SiJiMi. The full cDNA sequences of *MiCOL16A* and *MiCOL16B* were 1269 bp and 1251 bp, encoding 423 and 417 amino acids, respectively; the two genes were 73.06% identical. The DNA sequences of *MiCOL16A* and *MiCOL16B* were 2030 bp and 1751 bp, respectively, and each contained one intron (Figure 1A). The alignment of *MiCOL16A* and *MiCOL16B* indicated that both have one CCT and one B-box domain and are highly conserved with other genes (Figure 1B). Phylogenetic analysis showed that these two genes were highly identical to *Arabidopsis* AtCOL6 and AtCOL16 in group II (Figure 1C). Therefore, these results indicated that *MiCOL16A* and *MiCOL16B* belong to group II genes of the *CO* gene family.

### 2.2. Expression of MiCOL16A and MiCOL16B in Mango

For the tissue-specific expression tests, qRT–PCR was performed. The results showed that *MiCOL16A* and *MiCOL16B* were expressed in all the tested tissues. On different branches, the expression levels of *MiCOL16A* and *MiCOL16B* in the leaves were always higher than those in the stems, buds or flowers, and the lowest expression levels of *MiCOL16A* and *MiCOL16B* were detected in the buds and flowers. In contrast, the expression level of *MiCOL16A* in the leaves of nonflowering branches was higher than that in the leaves of flowering branches (Figure 2A), but *MiCOL16B* expression was lower in the leaves of nonflowering branches than in those of flowering branches (Figure 2B).

To analyze the temporal expression patterns of *MiCOL16A* and *MiCOL16B*, qRT–PCR was performed. The results suggested that the two target genes were expressed in the leaves of mango at all tested periods. The expression pattern of *MiCOL16A* gradually increased from vegetative growth to the late floral induction period and then decreased (Figure 2C). However, *MiCOL16B* gene expression increased from the vegetative growth period to the early floral induction period and then decreased (Figure 2D).

The circadian-driven expression of *MiCOL16A* and *MiCOL16B* was determined using total RNA isolated from ‘SiJiMi’ mango leaves, which were collected every 3 h for three days. The results suggested that under LD conditions, the expression level of *MiCOL16A* started to increase after dusk, peaked 6 h after dusk and decreased rapidly thereafter before beginning to increase again after dusk (Figure 3A). Interestingly, this pattern also appeared under MD and SD conditions, and this result proved that *MiCOL16A* expression may be induced by night treatment and is not affected by the length of light (Figure 3C,E). The expression level of *MiCOL16B* fluctuated with time under different conditions, but there was no regularity. These results suggest that *MiCOL16A* expression exhibits a diurnal oscillation rhythm, but *MiCOL16B* expression is not affected by a diurnal rhythm (Figure 3B,D,F).

### 2.3. Both MiCOL16A and MiCOL16B Are Nuclear Proteins with Transcriptional Activation Activity

*MiCOL16A* and *MiCOL16B* protein-linked GFP fusion constructs driven by the CaMV 35S promoter were developed for molecular function assays. The GFP fusion constructs were inserted into vectors, which were transferred into onion inner epidermal cells. Single-strand analysis indicated that free GFP localized to the nucleus and cytomembrane, and both the MiCOL16A and MiCOL16B proteins localized to the nucleus (Figure 4A).

A transcriptional activity assay was performed in yeast cells to demonstrate whether *MiCOL16A* or *MiCOL16B* had transcriptional activation activity. According to a previous study, the MR between the B-BOX and CCT domains is required for CO transcriptional activity [24,32,33]. Thus, *MiCOL16A*, *MiCOL16B*, *MiCOL16A*-ΔMR and MiCOL16B-ΔMR were fused into a pGBKT7 vector, and an empty pGBKT7 vector was used as a control. The five different plasmids were transferred into yeast cells, which were then transferred onto different plates. Three days later, all the transformants grew equally well on selective SD/-Trp media. On these three selective media, the BD-MiCOL16A and BD-MiCOL16B transformants grew well and turned blue, but the yeast transformed with the BD-vector, BD-MiCOL16A-ΔMR and BD-MiCOL16B-ΔMR grew only on SD/-Trp/X-α-gal plates and did not turn blue (Figure 4B). Together, α-gal activity could not be detected when the MR region was deleted. These results showed that through their MR domains, *MiCOL16A* and *MiCOL16B* have transcriptional activation activity in yeast.

### 2.4. Overexpression of MiCOL16A and MiCOL16B Delayed Flowering in Arabidopsis

To determine whether *MiCOL16A* and *MiCOL16B* are involved in the regulation of flowering time, they were overexpressed in *Arabidopsis* (under the control of the CaMV 35S promoter). We obtained 15 and 10 independent transgenic lines of *MiCOL16A* and *MiCOL16B*, respectively, and we selected three homozygous lines for each construct from within the T3 generation and planted them under LD or SD conditions. The PCR analysis results showed that *MiCOL16A* and *MiCOL16B* were expressed in the transgenic plants but not in the WT or empty vector-transformed *Arabidopsis* plants under LD (Figure 5(A1,B1)) or SD conditions (Figure 6(A1,B1)). Under LD and SD conditions, both the *MiCOL16A* and *MiCOL16B* transgenic lines flowered later than the WT and empty vector-transformed plants. At flowering, compared with the WT and empty vector-transformed plants, the *MiCOL16A* and *MiCOL16B* transgenic plants had more rosette leaves (Figure 5 and Figure 6) (Table 1).

To further dissect the expression patterns of the floral integrator genes in the *MiCOL16A* and *MiCOL16B* overexpression lines, the transcript levels of *AtFT* and *AtSOC1* were measured in the WT and overexpression plants under LD or SD conditions (Figure 7). The results showed that both *MiCOL16A* and *MiCOL16B* significantly repressed the expression of *AtSOC1* and *AtFT* in *Arabidopsis* under LD and SD conditions.

### 2.5. MiCOL16A and MiCOL16B Enhance Tolerance to Abiotic Stress

To assess the effect of ectopic *MiCOL16A* and *MiCOL16B* expression in response to abiotic stress, three homozygous lines (T3 generation) were selected. Three-day-old seedlings of the overexpression and WT plants were transplanted onto half-strength MS media supplemented with mannitol and NaCl. The length of their roots was measured after 7 days of stress treatment. The untreated WT and overexpression plants did not significantly differ, but compared with the WT plants, both *MiCOL16A* and *MiCOL16B* overexpression plants grew better and had longer roots under all stress conditions (Figure 8 and Figure 9). Together, these results showed that, compared with WT plants, *MiCOL16A* and *MiCOL16B* transgenic plants had improved tolerance to drought and salt stress.

To further determine the response of *MiCOL16A* and *MiCOL16B* transgenic plants to abiotic stress, 7-day-old seedlings were transplanted into square pots. After the seedlings were allowed to recover, they were watered with 300 mM NaCl every 2 days, and regular water was withheld (Figure 10A,B). For salt treatment, the survival rate was measured when obvious phenotypic differences occurred. Approximately 20.0% of WT plants, 80.0% of *OEMiCOL16A*#3 plants, 93.3% of *OEMiCOL16A*#6 plants and 86.7% of *OEMiCOL16A*#10 plants survived (Figure 10C). Similarly, approximately 26.7% of WT plants survived, but 73.3% of *OEMiCOL16B*#3, 93.3% of *OEMiCOL16B*#12 and 86.7% of *OEMiCOL16B*#13 plants survived (Figure 10D). With respect to the drought treatment, the survival rate was measured after the plants had been rewatered for 3 days: a total of 6.7% of WT plants survived, but 93.3% of *OEMiCOL16A*#3, 80.0% of *OEMiCOL16A*#6 and 73.3% of *OEMiCOL16A*#10 plants survived (Figure 10C). Similarly, no WT survived, but 40.0% of *OEMiCOL16B*#3, 93.3% of *OEMiCOL16B*#12 and 80.0% of *OEMiCOL16B*#13 lines survived (Figure 10D). Together, these results showed that overexpression of *MiCOL16A* and *MiCOL16B* in *Arabidopsis* improved salt and drought tolerance.

To explore the molecular mechanism underlying these phenomena in the transgenic lines in response to salt or drought, four stress-related genes were selected: *AtNHX1*, *AtRD20*, *AtSOS1* and *AtCOR15A* (Figure 11). Under salt stress, the expression levels of *AtNHX1*, *AtRD20* and *AtSOS1* were significantly higher in the three transgenic lines of *MiCOL16A* than in the WT, but in the *MiCOL16B* transgenic lines, the expression level of *AtRD20* was not significantly higher (Figure 11A,B). Under drought conditions, the expression levels of *AtCOR15A*, *AtRD20* and *AtNHX1* in the *MiCOL16A* and *MiCOL16B* transgenic lines were significantly higher than those in the WT (Figure 11C,D). These findings showed that by regulating the expression of stress-responsive genes, the *MiCOL16A* and *MiCOL16B* genes might increase the stress tolerance of transgenic plants under drought or salt stress.

## 3. Discussion

Flowering is an important event in the life cycle of plants. Mango, a typical perennial and economically important fruit tree species, is distributed mainly in tropical and subtropical areas, and its economic benefits are greatly affected by flowering time. Therefore, successfully controlling flowering time is critically important in mango production. The *CO* gene plays an indispensable role in the photoperiodic pathway and regulates the flowering time of plants [34]. In a previous study, more than 36 *CO* homologs were identified from mango transcriptomic data [27]. Here, we identified and characterized two zinc finger COL16 protein orthologs, *MiCOL16A* and *MiCOL16B*, in mango. Bioinformatic analysis indicated that both MiCOL16 proteins are highly homologous to AtCOL16 and have only one CCT and one B-box domain, thus belonging to group II of the *CO* gene family.

*CO/COL* genes are the key genes controlling flowering in plants. These genes are regulated by the upstream gene *GI* [23,35] and control the downstream gene *FT*, thereby inducing flowering in plants [36]. Studies have shown that the leaves are the plant organs that initially perceive photoperiodic signals [37], and the synthesis of FT proteins controlled by the *CO* gene is also observed to occur in the leaves [38,39]. In this study, the expression patterns of the *MiCOL16A* and *MiCOL16B* genes in SiJiMi showed that both genes were highly expressed in the leaves, especially during the vegetative phase, but were expressed at low levels in the flowers (Figure 2). Similarly, the *GbCOL16* gene in *Ginkgo biloba* is most highly expressed in the leaves [40], and the *BvCOL1* gene in *Beta vulgaris* is most highly expressed in cauline leaves and less so in the buds and roots [41]. In petunia, *PhCOL16* homologs are involved in chlorophyll accumulation and have higher expression levels in leaves [42]. Moreover, *OsCOL16* in rice is expressed mainly in the youngest leaf blade, and the *VviCOL16a* genes in *Vitis vinifera* were expressed the most in the leaves [24,43]. In contrast, the highest expression level of *MiCOL16A* was found in the leaves of nonflowering branches and during the late floral induction period, but the *MiCOL16B* gene was expressed mainly in the leaves of flowering branches and during the early floral induction period.

In general, *CO* genes are affected by the circadian rhythm. In *Arabidopsis*, the expression level of *AtCO* peaks at dusk under LD conditions but peaks at night under SD conditions [44]. In *Cymbidium goeringii*, *CgCOL* is expressed at higher levels in the light than in the dark under LD conditions but at lower levels in the light than in the dark under SD conditions [45]. In *Chrysanthemum morifolium*, CmCOL expression peaks at 4:00 and 16:00 under both SD and LD conditions. Moreover, the expression level of the *OsCOL16* gene in rice increases after decreasing under both SD and LD conditions but peaks at 6 h or 14 h after dusk [24]. In the present study, the expression of *MiCOL16A* increased in the dark and decreased in the light. However, *MiCO* gene expression was greatest at 9:00 but then decreased under normal light conditions [31]. This result may be related to the structural differences of the genes. In the present study, the *MiCOL16A* gene was found to be involved in the circadian rhythm, but the *MiCOL16B* gene did not exhibit any regularity. The difference in the expression levels of these two genes may be caused by homologous differences.

According to the results, the MiCOL16A and MiCOL16B proteins localized to the nucleus of onion cells, and both have transcriptional activation activity through their MR domains. In rice, *OsCOL10*, *OsCOL15* and *OsCOL16* have one CCT and one B-box domain; they localize to the nucleus and display transcriptional activity through their MR domain [24,32,33]. Moreover, the protein encoded by *AtCOL*7 in Arabidopsis, which has one B-BOX and one CCT domain, has been reported to be the key segment exhibiting transcriptional activity between the B-box and CCT domains [46]. The results in this report also confirmed the results of the present experiment.

*CO* acts at the center of the coordinate input mechanism that responds to light. The flowering time-controlling molecular mechanisms in *Arabidopsis* and rice have been studied extensively [47]. In *Arabidopsis*, the *CO* gene promotes flowering under LD conditions but inhibits flowering under SD conditions [8], and overexpression of the *COL16* gene can lead to slightly early flowering under LD conditions [20]. In addition, the *COL4* gene represses flowering under LD or SD conditions [48]; the structure of both *COL7* and *COL8* is similar to that of *MiCOL16*, and these genes delay flowering under LD conditions [21,22]. In rice, *OsHd1* represses flowering only under LD conditions [8], *OsCO3* delays flowering time under SD conditions [49], and *OsCOL16* inhibits flowering under both SD and LD conditions [24]. In the present experiment, both *MiCOL16A* and *MiCOL16B* inhibited the flowering of the transgenic plants under SD and LD conditions. These results are similar to those for the *OsCOL16* gene in rice.

*CO* controls flowering by inducing the expression of the downstream gene *FT* [36,50]. In *Arabidopsis*, *CO* promotes flowering by upregulating the expression of FT genes under LD conditions and delays flowering by decreasing *FT* gene expression [51]. *CO* homologs, such as the *COL8* and *COL9* genes, repress FT expression and repress flowering only under LD conditions [22,52]. In bamboo, *PvCO1* decreases the expression of *FT* to control flowering time [53]. The *OsCOL9* gene delays flowering by repressing the downstream flowering-promoting gene *Ehd1* [54]. In the present experiment, both the *MiCOL16A* and *MiCOL16B* genes repressed flowering by decreasing the expression of *AtSOC1* and *AtFT* under SD and LD conditions (Figure 7). Similarly, the *MiCO* gene also represses flowering by decreasing *AtFT* and *AtSOC1* expression [31]. Together, our findings suggest that the *MiCOL16A* and *MiCOL16B* genes regulate flowering time by affecting the expression levels of *SOC1* and *FT*.

The molecular responses of plants to abiotic stresses are complex and dynamic [55,56], and they involve interactions with many molecular pathways [57]. Similarly, plant responses to these stresses are also complex [58]. *CO* is universally known as a photoperiod-responsive gene [5]. In recent years, some experiments have indicated that *CO* and *CO* homologs have a certain impact on the responses of plants to abiotic stresses. In *Arabidopsis*, the expression of the *AtCOL4* gene is significantly upregulated under high-salt and osmotic stress, and the survival rate of *AtCOL4* transgenic plants is higher than that of *atcol4* and WT plants [59]. The expression level of *GhCOL16*, a *CO* homolog in cotton, was also upregulated 12 h after NaCl or polyethylene glycol (PEG) 6000 treatment [60]. In the present study, the lengths of the roots and survival rates of the *MiCOL16A* and *MiCOL16B* transgenic plants were all greater than those of the WT plants under salt or drought conditions. This result shows that *MiCOL16A* and *MiCOL16B* enhance the tolerance of the transgenic plants to abiotic stress. However, *BnCOL2* increases sensitivity to drought conditions in *Arabidopsis* plants, and *BnCOL2* overexpression results in a survival rate that is lower than that of the WT plants [61]. These results are different from the results of the present experiment and may be caused by the presence of different domains.

To some extent, the transcript levels of stress-related genes also indicate the tolerance of plants to abiotic stresses [62]. For example, stress-responsive gene expression was reduced, resulting in a decrease in the stress tolerance of *BnCOL2* transgenic lines [61]. In our study, all of the *MiCOL16A* and *MiCOL16B* transgenic plants were tolerant to drought and salt stress; the expression levels of *AtNHX1*, *AtRD20* and *AtSOS1* increased under salt treatment; and the expression levels of *AtCOR15A*, *AtRD20* and *AtNHX1* increased under drought treatment. *AtNHX1* and *AtRD20* play vital roles in the response to salt and drought stress [63,64,65], *AtSOS1* transcript levels substantially increase upon NaCl treatment [66], and the *AtCOR15A* gene is known as a marker of drought stress [67]. In summary, it can be hypothesized that by regulating stress-responsive genes, the *MiCOL16A* and *MiCOL16B* genes positively regulate the salt and drought stress tolerance of *Arabidopsis*.

## 4. Material and Methods

### 4.1. Materials

The samples (*M. indica* L. cv. SiJiMi) were collected from an orchard at Guangxi University, Nanning, Guangxi, China. The leaves, buds and stems of nonflowering and flowering branches were collected on 4 January 2019. Leaves were collected on 5 November 2018, 5 December 2018, 4 January 2019, 29 January 2019 and 6 March 2019. Leaves were collected every three hours on three separate days from 2-year-old mango trees grown under different conditions for diurnal rhythmic expression analysis (LD (16 h light/8 h dark), MD (12 h light/12 h dark) and SD (8 h light/16 h dark)). The plant samples were frozen in liquid nitrogen and kept at −80 °C. Wild-type (WT) *Arabidopsis* ecotype Columbia (Col-0) plants were used to study gene function and were grown in the laboratory.

### 4.2. Identification and Sequence Analysis

Total RNA of SiJiMi mango plants was extracted by using an RNAprep Pure Kit (for polysaccharide- and polyphenolic-rich plants, DP441) (Tiangen, Beijing, China). cDNA was reverse transcribed with 1 µg of total RNA with PrimeScript™ Reverse Transcriptase M-MLV (Takara, Dalian, China) and Oligo(dT)18 primers (TTTTTTTTTT TTTTTTTTTTTT). The cetyl-trimethylammonium bromide (CTAB) method was used to isolate genomic DNA from mango leaves [68]. The *MiCOL16* gene was isolated from the transcriptomic data of SiJiMi mango (unpublished data), after which the sequence was verified. The primers designed for cloning included SCOL16Au (5′-GCCTTTGCAA AATGATCACCGG-3′), SCOL16Ad (5′-GCGCCATTTATTTCTTGAGG-3′), SCOL16Bu (5′-GTCTTTGCGGTATGATCACTG-3′) and SCOL16Bd (5′-CGGCCATCACCATTTAT TTC-3′). The PCR conditions were as follows: 95 °C for 3 min; 38 cycles of 95 °C for 30 s, 53 °C/52 °C for 30 s and 72 °C for 1 min 30 s; and 72 °C for 10 min. The final products were inserted into a pMD18-T vector and sequenced.

The amino acid sequences of *MiCOL16A* and *MiCOL16B* were analyzed with the BLAST search tool (https://www.ncbi.nlm.nih.gov/BLAST/, accessed on 20 May 2019). *CO* homologous sequences were downloaded from the NCBI database (https://www.ncbi.nlm.nih.gov, accessed on 25 May 2019). Sequence alignment was subsequently performed using DNAMAN 7 software (Lynnon Corporation, Pointe-Claire, QC, Canada). A neighbor-joining tree based on the *CO* gene family in *Arabidopsis* and several other tree species was constructed via MEGA 6.06, with 1000 bootstrap replicates.

### 4.3. qRT–PCR Analysis

First-strand cDNA of each sample was synthesized, and *MiActin1* (qMiACTu, 5′-CCGAGACATGAAGGAGAAGC-3′; qMiACTd, 5′-GTGGTCTCATGGATACCAGCA- 3′) was used as an internal control gene for data processing [69]. The gene-specific primers used for *MiCOL16A* and *MiCOL16B* were qCOL16Au (5′-TGAATCACCACTG GCAGCTGA-3′) and qCOL16Ad (5′-GGGGTTCCTGTTGTCCATGGA-3′) as well as qCOL16Bu (5′-ACTTCGGAGACAGCACAGTGA-3′) and qCOL16Bd (5′-TTGCATTCGG TGTTCGCCATT-3′). qRT–PCR was performed with an ABI 7500 Real-Time PCR System (Applied Biosystems, Foster City, CA, USA) in conjunction with SYBR Premix Ex Taq II (Takara, Dalian, China). The expression data were normalized according to the 2^−ΔΔCt^ method [70].

### 4.4. Subcellular Localization

The *MiCOL16A* and *MiCOL16B* coding DNA sequence (CDS) regions were cloned into a green fluorescent protein (GFP)-encoding gene via the cauliflower mosaic virus (CaMV) 35S promoter. GFP fusions were transferred into *Agrobacterium tumefaciens* EH105, which were then transiently injected into onion inner epidermal cells. Fluorescent cells were observed via laser confocal microscopy. An empty vector with only *GFP* was used as the control, and 4′,6-diamidino-2-phenylindole (DAPI) was used to visualize the location of the nucleus.

### 4.5. Analysis of Transcriptional Activity of MiCOL16A and MiCOL16B

To determine the transcriptional activation domains, BD-MiCOL16A, BD-MiCOL16B, BD-MiCOL16A-ΔMR and BD-MiCOL16B-ΔMR constructs were generated, and the full-length CDS and middle region (MR) deletion of *MiCOL16A* and *MiCOL16B* were amplified via PCR and inserted into the pGBKT7 expression vector (Clontech, Dalian, China). An empty pGBKT7 vector was used as a control (BD-vector). These four plasmids and empty pGBKT7 vectors were subsequently transferred into Y2H Gold yeast cells. The seedlings were subsequently grown on sucrose–dextran (SD) media lacking tryptophan for 3 days. The yeast strains that grew successfully were then transferred onto SD/-Trp, SD/-Trp/X-α-gal and SD/-Trp/X-α-gal/AbA media and grown for another 3 days.

### 4.6. Vector Construction and Arabidopsis Plant Transformation

The complete CDSs of *MiCOL16A* and *MiCOL16B* were amplified and inserted into a pBI121 vector. The gene-specific primers used were as follows: ZCOL16Au (5′-TCTAGAATGATCACCGGAAAG-3′; the *Xba* I site is underlined), ZCOL16Ad (5′-CCCGGGTTATTTCTTGAGGTAAGG-3′; the *Xma* I site is underlined), ZCOL16Bu (5′-GCGGTTCTAGAATGATCTCTG-3′; the *Xba* I site is underlined) and ZCOL16Bd (5′-CCCGGGTTATTTCTTCAGGTAGGG-3′; the *Xma* I site is underlined). The pBI121 vector was subsequently digested by *Xba* I and *Xma* I restriction enzymes. Positive recombinant pBI121-MiCOL16A and pBI121-MiCOL16B plasmids were then transformed into *Agrobacterium tumefaciens* EH105. The primers JCOL16Au (5′-GCCTTTGCAAAATGATCACCGG-3′), JCOL16Bu (5′-GTCTTTGCGGTATGATCAC TG-3′) and 121GUSd (5′-TTGGGACAACTCCAGTGAAAAG-3′) were used to assess the target bacterial colonies. *Arabidopsis* plants were subsequently transformed with the floral-dip method [71]. Transgenic *Arabidopsis* plants growing on half-strength Murashige and Skoog (MS) media supplemented with 50 mg/L kanamycin were identified and then transplanted into square pots filled with a mixture of organic substrate, peat moss and vermiculite (2:2:1), after which they were allowed to grow unabated for 7 days. Seeds of T3-generation homozygous plants were sown at 22 °C under SD conditions (8 h light/16 h darkness) or LD conditions (16 h light/8 h darkness) to measure flowering time, plant height and rosette leaves. Two-week-old WT and transgenic plants were sampled for qRT–PCR analysis. The flowering-related genes *AtSOC1* (No. AY007726; primers F, 5′-CGAGCAAGAAAGACTCAAGTGTTTAAGG-3′; R, 5′-TTCATGAGATCCCCA CTTTTCAGAGAG-3′) and *AtFT* (No. AB027504; primers, 5′-CTTGGCAGGCAAACAGTGTATGCAC-3′; R, 5′-GCCACTCTCCCTCTGACAATTG TAGA-3′) were used for qRT–PCR analysis.

### 4.7. Stress Treatments of Transgenic and WT Plants

Seeds of the transgenic lines and WT were sown evenly on the surface of half-strength MS media without antibiotics and stored at 4 °C for 3 days for vernalization.

For root growth assays, WT and *MiCOL16A* and *MiCOL16B* transgenic lines were subjected to stress, and 3 days later, all the plants were transferred to vertically oriented square-shaped containers filled with half-strength MS agar media supplemented with NaCl (0, 100 and 200 mM) or mannitol (0, 300 and 500 mM) [72,73]. The plants were then grown upright in an artificial climate chamber under LD conditions at 22 °C. Seven days later, the length of the primary roots of all lines was measured.

To assess the stress tolerance of transgenic *Arabidopsis*, 7-day-old seedlings were transplanted into square pots. For salt stress, the WT and transgenic lines were treated with 300 mM NaCl solution every 2 days until obvious phenotypic differences occurred, and then the survival rate was determined. For drought treatment, water was withheld from the transgenic lines and WT plants until obvious phenotypic differences occurred. After rewatering for 3 days, the survival rates of the overexpression lines and WT plants were measured.

To investigate the expression patterns of related genes in transgenic *Arabidopsis* plants in response to stress, seeds of the transgenic lines and WT were sown on half-strength MS media supplemented with NaCl or mannitol. Fifteen days later, total RNA was extracted. The specific primers used for the stress-related genes *AtNHX1* (No. AT5G27150; F, 5′-AGCCTTCAGGGAACCACAAT-3′; R, 5′-CTCCAAAGACGGGTCGC ATG-3′) [64], *AtRD20* (No. AT2G33380; F, 5′-ATCGACAACATACACAAAGCCAA-3′; R, 5′-TCCATCAA AGCAACCTCTCACAG-3′) [63], *AtSOS1* (No. AT2G01980; F, 5′-AGTGTAAGTTTCGGTGGGATC-3′; R, 5′-CACGCATGTTTACGGGTTTC-3′) [74], and *AtCOR15A* (No. AT2G42540; F, 5′-TTCCACAGCGGAGCCAAGCA-3′; R, 5′-AGCGGCGTAGATCAACGACTTC-3′) [67] were used to perform qRT–PCR, and *AtActin2* (No. At3g18780; F, 5′-CACTGTGCCAATCTACGAGGGT-3′; R, 5′-GCTGGAA TGTGCTGAGGGAAG-3′) was used as a reference control.

### 4.8. Statistical Analysis

All experiments were repeated at least three times. The standard deviations (SDs) are represented by error bars in the figures. All statistical analyses were performed by SPSS 17.0, and significance was tested at the *p* < 0.05 (* in the figures) and *p* < 0.01 (** or Lowercase letters in the figures) levels. Bar charts were generated with GraphPad Prism 7 software.

## 5. Conclusions

In this study, we isolated the *MiCOL16A* and *MiCOL16B* genes from SiJiMi mango. Both the *MiCOL16A* and *MiCOL16B* proteins localize to the nucleus and have transcriptional activity through their MR domain. Overexpression of *MiCOL16A* and *MiCOL16B* repressed flowering in *Arabidopsis* under SD and LD conditions and improved tolerance to drought and salt stress conditions. Overall, our results clearly demonstrated that the *MiCOL16A* and *MiCOL16B* genes play key roles in mango flowering and abiotic stress responses.

## Figures and Tables

**Figure 1 ijms-23-03075-f001:**
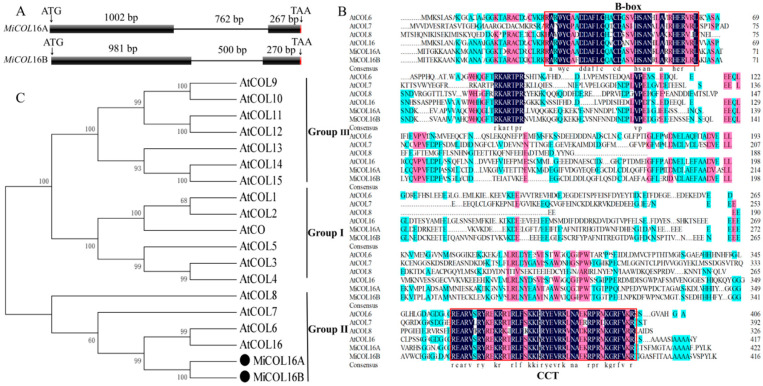
Sequence analysis of *MiCOL16A* and *MiCOL16B*. (**A**) Genomic structures of *MiCOL16A* and *MiCOL16B*. Exons are represented by black squares, and introns are represented by black lines. The numbers indicate the lengths of the corresponding regions. (**B**) Alignment of the predicted amino acid sequences of AtCOL6–8 and AtCOL16 in *Arabidopsis* and of MiCOL16A and MiCOL16B in mango. The conserved CCT and B-box domain regions are indicated with red boxes. The dark color indicates that the amino acids are 100% conserved. The genes and their accession numbers are shown in Appendix A. (**C**) Phylogenetic relationship of CO/COL proteins in *Arabidopsis* and mango. MiCOL16A and MiCOL16B are represented by black solid circles.

**Figure 2 ijms-23-03075-f002:**
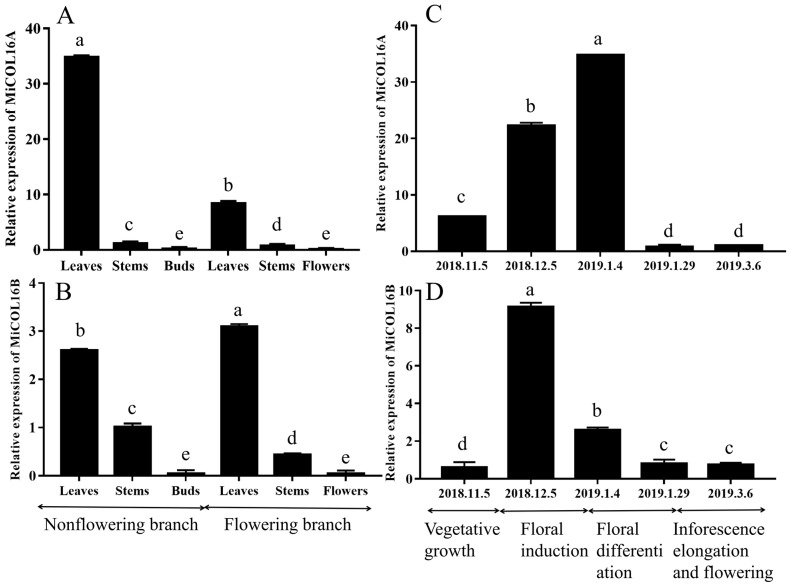
Tissue-specific and temporal expression analysis of the *MiCOL16A* and *MiCOL16B* genes. (**A**,**B**) Expression patterns of the *MiCOL16A* and *MiCOL16B* genes in different tissues of SiJiMi mango. (**C**,**D**) Expression patterns of the *MiCOL16A* and *MiCOL16B* genes at different time points.

**Figure 3 ijms-23-03075-f003:**
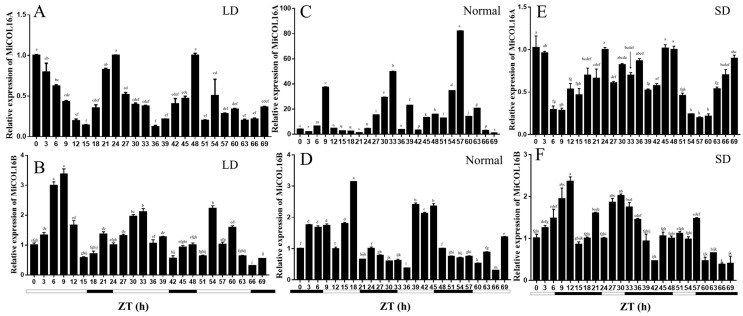
Expression analysis of the *MiCOL16A* and *MiCOL16B* genes in terms of circadian rhythm. (**A**,**C**,**E**) Expression patterns of *MiCOL16A* under LD (**A**), normal (**C**) and SD (**E**) conditions. (**B**,**D**,**F**) Expression patterns of *MiCOL16**B* under LD (**B**), normal (**D**) and SD (**F**) conditions.

**Figure 4 ijms-23-03075-f004:**
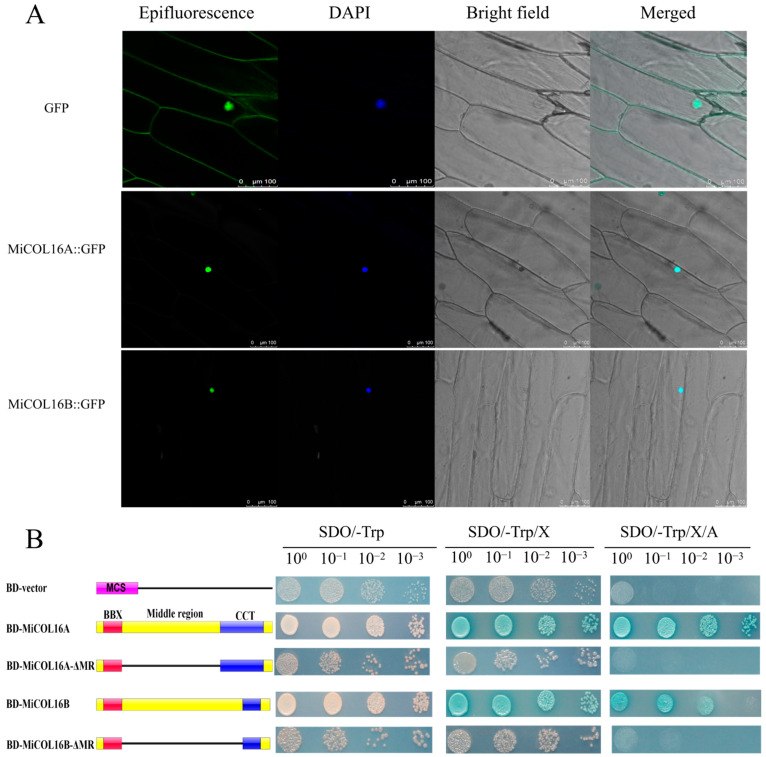
(**A**) Subcellular localization. Scale bars = 100 µm. (**B**) Transcriptional activation activity. The left diagram shows various constructs. MCS, multiple cloning sites; BD, GAL4-DNA binding domain; BBX, B-box; CCT, CO, CO-like, TOC1; MR, middle region.

**Figure 5 ijms-23-03075-f005:**
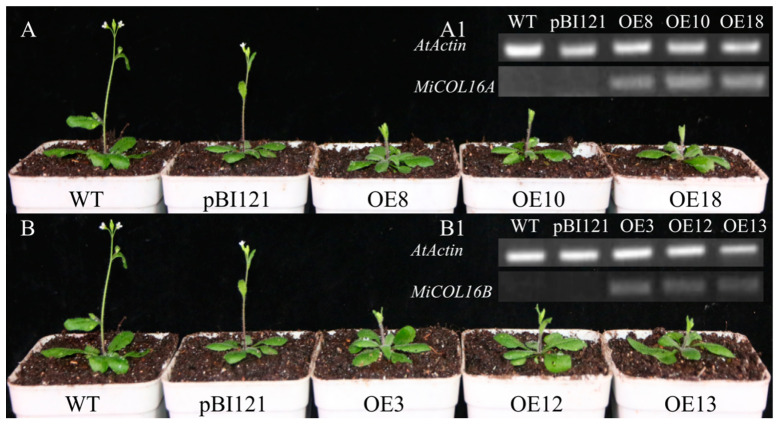
Ectopic expression of *MiCOL16A* and *MiCOL16B* delayed flowering under LD conditions. (**A1**,**B1**) Expression of *MiCOL16A* (**A1**) and *MiCOL16B* (**B1**) in the WT, empty vector-transformed and overexpression plants. (**A**) *MiCOL16A* transgenic *Arabidopsis* plants under LD conditions. (**B**) *MiCOL16B* transgenic *Arabidopsis* plants under LD conditions.

**Figure 6 ijms-23-03075-f006:**
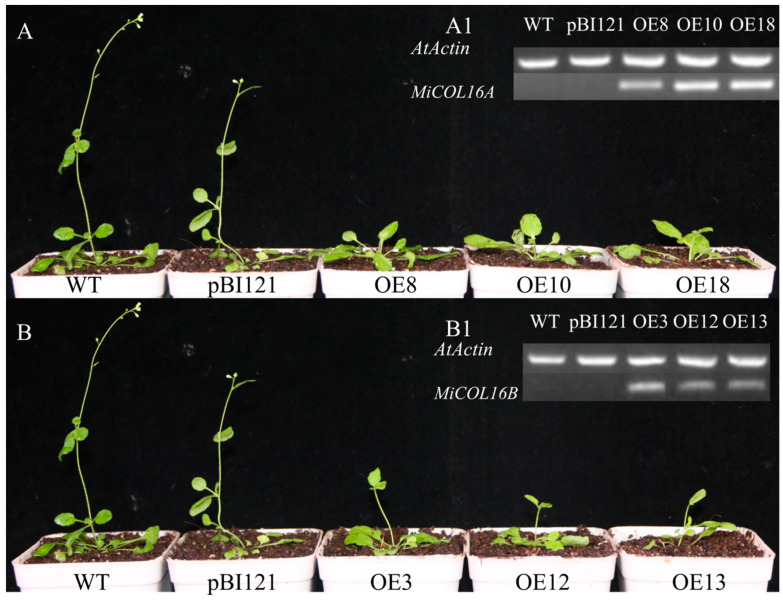
Ectopic expression of *MiCOL16A* and *MiCOL16B* delayed flowering under SD conditions. (**A1**,**B1**) Expression of *MiCOL16A* (**A1**) and *MiCOL16B* (**B1**) in the WT, empty vector-transformed and overexpression plants. (**A**) *MiCOL16A* transgenic *Arabidopsis* plants under SD conditions. (**B**) *MiCOL16B* transgenic *Arabidopsis* plants under SD conditions.

**Figure 7 ijms-23-03075-f007:**
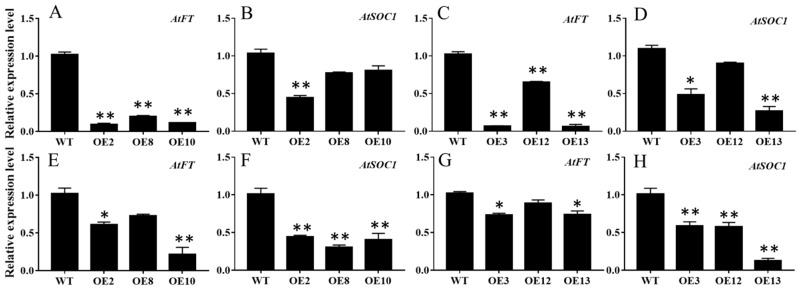
Expression patterns of flowering-related genes. (**A**–**D**) Expression patterns of *AtFT* and *AtSOC1* in the WT and the *MiCOL16A* (**A**,**B**) or *MiCOL16B* (**C**,**D**) overexpression plants under LD conditions. (**E**–**H**) Expression levels of *AtFT* and *AtSOC1* in the WT and the *MiCOL16A* (**E**,**F**) or *MiCOL16B* (**G**,**H**) overexpression plants under SD conditions. Significant differences among the samples were assessed at the *p* < 0.05 (*) and *p* < 0.01 (**) levels by Student’s *t* tests.

**Figure 8 ijms-23-03075-f008:**
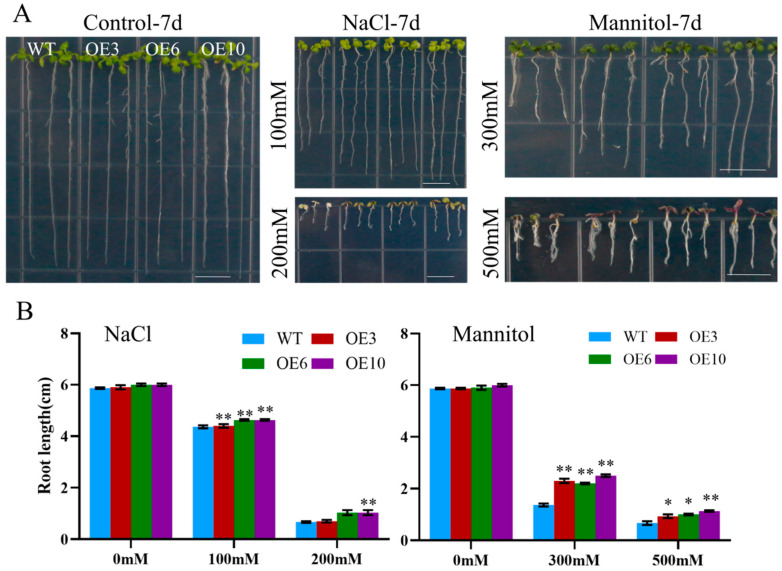
Assays of the length of the primary roots of WT and *MiCOL16A* transgenic lines under abiotic stress. (**A**) Seeds of the WT and three transgenic lines grown on half-strength MS media and subjected to various stresses. The bars represent 1.0 cm. (**B**) Lengths of the roots of all the lines under salt and drought treatment. Significant differences among the samples were assessed at the *p* < 0.05 (*) and *p* < 0.01 (**) levels by Student’s *t* tests.

**Figure 9 ijms-23-03075-f009:**
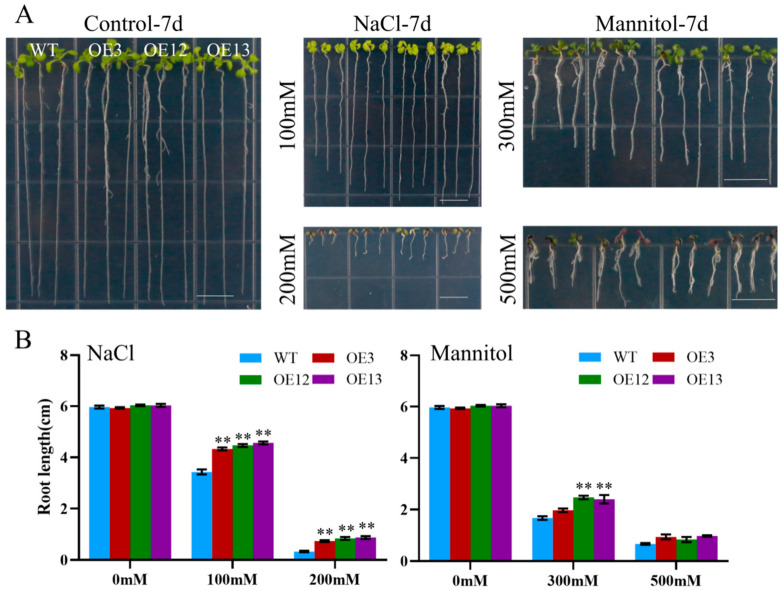
Assays of the length of the primary roots of the WT and *MiCOL16B* transgenic lines under abiotic stress. (**A**) Seeds of the WT and three transgenic lines grown on half-strength MS media and subjected to various stresses. The bars represent 1.0 cm. (**B**) Lengths of the roots of all the lines under salt and drought treatment. Significant differences among the samples were assessed at the *p* < 0.01 (**) levels by Student’s *t* tests.

**Figure 10 ijms-23-03075-f010:**
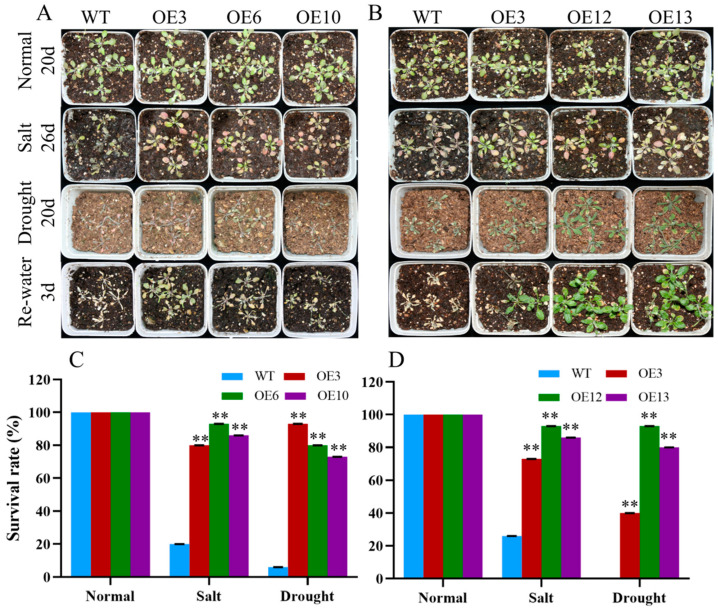
Phenotypes of WT and *MiCOL16A* and *MiCOL16B* transgenic plants under abiotic stresses. Normal, control; salt, 300 mM NaCl solution applied every 2 days; drought, withholding of water. (**A**) WT and *MiCOL16A* transgenic plants under different stresses. (**B**) WT and *MiCOL16B* transgenic lines under different stresses. (**C**–**D**) Survival rates of the WT and the *MiCOL16A* (**C**) and *MiCOL16B* (**D**) transgenic lines under different stresses. Significant differences among the samples were assessed at the *p* < 0.01 (**) levels by Student’s *t* tests.

**Figure 11 ijms-23-03075-f011:**
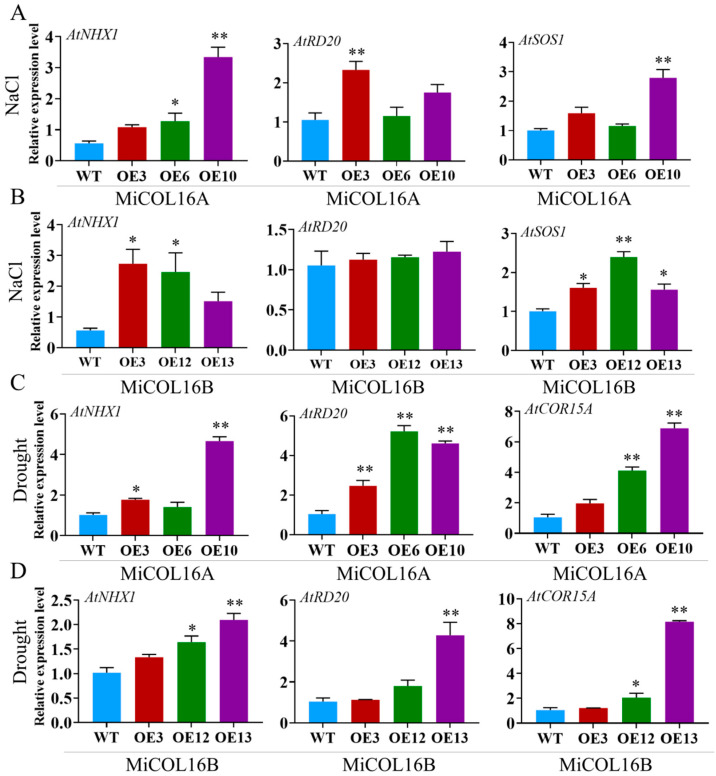
Expression patterns of stress-responsive genes in WT and *MiCOL16A* and *MiCOL16B* transgenic lines. (**A**,**B**) Salt stress conditions. Expression levels of the *AtNHX1*, *AtRD20* and *AtSOS1* genes in *MiCOL16A* (**A**) and *MiCOL16B* (**B**) transgenic lines. (**C**,**D**) Drought stress conditions. Expression levels of the *AtNHX1*, *AtRD20* and *AtCOR15A* genes in the *MiCOL16A* (**C**) and *MiCOL16B* (**D**) transgenic lines. Significant differences among the samples were assessed at the *p* < 0.05 (*) and *p* < 0.01 (**) levels by Student’s *t* tests.

**Table 1 ijms-23-03075-t001:** Overexpression of *MiCOL16A* and *MiCOL16B* repressed flowering in *Arabidopsis*.

ID	Days to Flowering		No. Rosette Leaves		Plant Height ^a^ (cm)	
	LD	SD	LD	SD	LD	SD
WT	24.8 ± 0.6	50.1 ± 0.8	5.5 ± 0.5	7.2 ± 0.3	6.4 ± 0.8	11.4 ± 0.6
pBI121	25.6 ± 0.4	49.6 ± 0.7	7.6 ±0.6	7.0 ± 0.5	6.7 ± 0.4	11.7 ± 0.4
*MiCOL*16A						
OE2	28.0 ± 0.3 *	51.3 ± 0.8	6.3 ± 0.4	7.6 ± 0.3	4.6 ± 1.1 *	14.2 ± 0.9 *
OE8	28.0 ± 0.5 *	52.8 ± 0.2 *	6.8 ± 0.6 *	7.3 ± 0.7	4.5 ± 0.6 *	14.2 ± 0.8 *
OE10	28.1 ± 0.8 *	52.8 ± 0.3 *	7.5 ± 1.0 *	7.3 ± 0.5	6.2 ± 1.1	13.0 ± 0.3 *
*MiCOL*16B						
OE3	27.6 ± 0.5 *	53.0 ± 0.4 *	6.6 ± 0.7 *	7.4 ± 0.6	4.7 ± 1.0 *	17.7 ± 1.1 *
OE12	26.9 ± 0.8 *	51.2 ± 0.9	7.1 ± 0.5 *	7.3 ± 0.4	5.2 ± 1.1 *	17.1 ± 1.0 *
OE13	28.4 ± 0.5 *	51.5 ± 0.4 *	6.7 ± 0.7 *	6.6 ± 0.6	4.3 ± 0.8 *	13.4 ± 0.2 *

^a^ Plant height was measured at the time of flowering. Significant differences among the samples were assessed at the *p* < 0.05 (*) level by Student’s *t* tests.

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
