# Peer review of "Isolation and Functional Characterization of Two CONSTANS-like 16 (MiCOL16) Genes from Mango"

_ijms, 2022, doi:10.3390/ijms23063075_

Round 1

Reviewer 1 Report

This manuscript present the isolation and functional analyses of two genes important for flowering regulation in mango. Overall, this study is well designed and organized. The writing is pretty clear to me. The conclusions can be strongly supported by the solid experimental and analysis results. This work will benefit our understanding of the gene network of flowering control genes in mango. The manuscript can be improved if the small concerns below can be addressed.

  1. In Figure 3, the y-axis labels are the same to Figure 2, and should be given for each panel.
  2. In Figure 7, the position of panels should be consistent across all figures of the whole manuscript, such as Top left.
  3. In Figures 7, 8B, 9B, 10CD, 11, the statistical test method and the corresponding p value should be provided in the legend if applied.

Author Response

Re:All revised text is marked in red font in manuscript.

In Figure 3, the y-axis labels are the same to Figure 2, and should be given for each panel.

Re:We have revised it in Figure 2 and Figure 3.

In Figure 7, the position of panels should be consistent across all figures of the whole manuscript, such as Top left.

Re:We have revised it in Figure 7.

In Figures 7, 8B, 9B, 10CD, 11, the statistical test method and the corresponding p value should be provided in the legend if applied.

Re:We have revised it in Figure 7, 8B, 9B, 10CD, 11.

Reviewer 2 Report

Flowering is very important stage in plant growth of each plant. Successfully controlling flowering period in mango is specially important for its final production. In the presented study two CONSTANS homologs, MiCOL16A and MiCOL16B, were isolated from the “SiJiMi” mango to elucidate the mechanisms controlling mango flowering. Research in this area is important (74 references are listed in the manuscript) and can have practical implications in relation to the growing conditions. The scientific article has a standard structure with all necessary chapters including conclusions. I consider the conclusions in the manuscript are original and bring some new interesting finding and information as the results of research demonstrated that MiCOL16A and MiCOL16B not only regulate flowering but also play a role in the abiotic stress response in mango. I did not find any serious defects in the work or in the presentation or ethical problems. In my opinion, the keywords are consistent with the content of the article. In conclusion, I can state that after studying the article, I do not have any serious comments and manuscript “Isolation and functional characterization of two CONSTANS- Like 16 (MiCOL16) genes from mango“meets the requirements and can be published in a IJMS.

Author Response

Thanks for your positive comments.

Reviewer 3 Report

The authors of this study have examined the role of two isoforms of constans. They have checked the temporal expression of these genes and its subcellular localization. Moreover, authors have identified the role of these in regulation of flowering and response to abiotic stress response. The experiments are planned well. I recommend this article for publication.

Author Response

We agree with and accept the suggestions of the editor and reviewer。

We have further optimized the manuscript and deeply polished the language.